# Explainable Molecular Property Prediction: Aligning Chemical Concepts with Predictions via Language Models

## Abstract

Providing explainable molecular property predictions is critical for many scientific domains, such as drug discovery and material science. Though transformer-based language models have shown great potential in accurate molecular property prediction, they neither provide chemically meaningful explanations nor faithfully reveal the molecular structure-property relationships. In this work, we develop a framework for *explainable mol*ecular property prediction based on *la*nguage models, dubbed as *Lamole*, which can provide chemical concepts-aligned explanations [1]. We take a string-based molecular representation — Group SELFIES — as input tokens to pre-train and fine-tune our *Lamole*, as it provides chemically meaningful semantics. By disentangling the information flows of *Lamole*, we propose considering both self-attention weights and gradients for better quantification of each chemically meaningful substructure's impact on the model's output. To make the explanations more faithfully respect the structure-property relationship, we then carefully craft a marginal loss to explicitly optimize the explanations to be able to align with the chemists' annotations. We bridge the manifold hypothesis with the elaborated marginal loss to prove that the loss can align the explanations with the tangent space of the data manifold, leading to concept-aligned explanations. Experimental results over six mutagenicity datasets and one hepatotoxicity dataset demonstrate *Lamole* can achieve comparable classification accuracy and boost the explanation accuracy by up to $14.3\%$, being the state-of-the-art in explainable molecular property prediction. The code is available at the provided link: `https://anonymous.4open.science/r/Lamole-7482`

## 1 Introduction

Molecular property prediction aims to reveal the molecular structures-property relationships, assisting scientists in screening molecules for various applications such as drug discovery and material design (Fang et al., 2022; Deng et al., 2023; Tripp et al., 2023; Wang et al., 2024; Ekström Kelvinius et al., 2023; Hong et al., 2024). Several learning-based models are devised based on the underlying molecular representations, such as graph-based and string-based molecular representations. Among them, string-based molecular representations, e.g., simplified molecular input line entry systems (SMILES (Weininger, 1988)), stand out for their simplicity and adaptability (Deng et al., 2023; Wigh et al., 2022; Cheng et al., 2023). By viewing the string-based molecular representation as a form of "chemical" language, the Transformer-based language models (LMs) like Bert (Kenton & Toutanova, 2019) offer higher throughput and accuracy for molecular property prediction (Deng et al., 2023; Chithrananda et al., 2020).

Despite the superior performance of learning-based prediction methods, *what key factors induce the model's predictions remain largely unexplored*, impeding further advancements in the scientific domains. Typically, it is crucial to obtain explanations of predictions while achieving accurate predictions. These obtained explanations could be used for scientific hypotheses validation or/and providing actionable insights for refining investigations, such as optimization for molecular structural design (Wu et al., 2023; Wellawatte et al., 2023; Das et al., 2022). With different types of molecular

---

[1]Lamole is from the name of a historical winery in Italy called Lamole di Lamole.

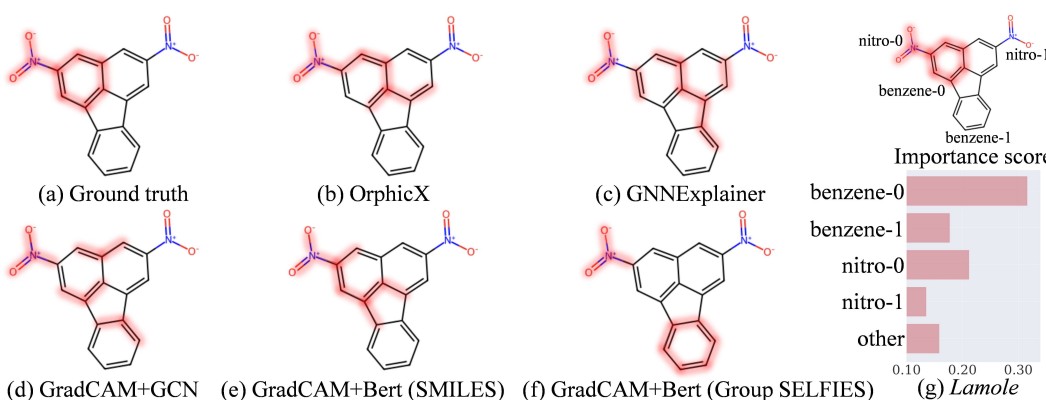

(a) Ground truth     (b) OrphicX     (c) GNNExplainer

(d) GradCAM+GCN   (e) GradCAM+Bert (SMILES)   (f) GradCAM+Bert (Group SELFIES)    (g) *Lamole*

Figure 1: (a) The molecule visualization of prediction/explanation. The interaction between the benzene ring and the nitro group (highlighted in red) induces the mutagenic property of the molecule. (b)-(e) are the explanation results obtained with various methods: (b) OrphicX (Lin et al., 2022); (c) GNNExplainer (Ying et al., 2019), (d) GNN with gradient-based explainability technique (Grad-CAM (Selvaraju et al., 2017)); (e) Bert with GradCAM (molecular string SMILES as input); (f) Bert with GradCAM (molecular string Group SELFIES (Cheng et al., 2023) as the input representation); (g) Our method *Lamole* assigns an importance score to each functional group/fragment to indicate their contribution to the property.

representations, explainability techniques for graph neural networks (GNNs) or LMs might be adopted to alleviate the general lack of explainability in molecular prediction (Proietti et al., 2024; Ying et al., 2019; Ye et al., 2023; Lin et al., 2021).

However, we argue that existing explainability techniques often struggle to generate plausible explanations that can highlight chemically meaningful substructures and faithfully uncover the structure-property relationships simultaneously. Specifically, 1) from the molecular representation perspective, the commonly used representations do not explicitly encode the chemically meaningful substructures; current explainability methods can only highlight individual atoms and bonds as explanations (see Fig. 1 (b)∼(e)). 2) From the perspective of explainability techniques, current methods suffer from two main limitations. First, they cannot effectively capture the interactions between functional groups within the molecular structure. Second, they could not generate explanations that align with chemists' intuition. As a result, they fail to produce explanations that faithfully reflect the structure-property relationships. (see Fig. 1 (f)). Therefore, an effective framework is imperative for explainable molecule property predictions.

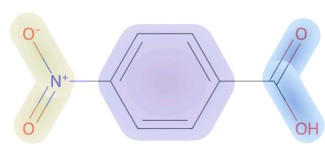

SMILES: c1cc(ccc1C(=O)O)[N+](=O)[O-]

Group SELFIES: [:0benzene][Ring2] [:0nitro][pop][Branch][:1carboxyl][pop]

Figure 2: The SMILES and Group SELFIES strings of p-nitrobenzoic acid molecule ($C_7H_5NO_4$): The tokens in the Group SELFIES string highlighted by color are the corresponding functional groups.

The recently proposed string-based molecular representation — Group SELFIES (Cheng et al., 2023) — encodes molecules at the functional group/fragment level, showcasing the possibility of obtaining chemically meaningful explanations. As shown in Fig. 2, Group SELFIES converts a p-nitrobenzoic acid molecule to a string, which explicitly encodes chemically meaningful substructures as tokens, including a benzene, a nitro group, and a carboxyl group. Compared with 2D molecular graphs, Group SELFIES provides inherent semantic information, making it easier for the model to capture and understand chemically meaningful semantics. Moreover, using Group SELFIES eliminates the need to identify or segment chemically meaningful substructures in 2D molecular graphs. With Group SELFIES's simplicity and adaptability, this work develops an *ex*plainable *mol*ecular property prediction framework based on *la*nguage models to provide chemical concepts-aligned explanations, called *Lamole* (see Fig. 1 (g)). The contributions can be summarized as follows.

1. We found that existing explainability techniques fail to provide chemically meaningful explanations, perceive functional group interactions, and reveal molecular structure-property relationships faithfully. To address the issues, we use Group SELFIES to pre-train and fine-tune LMs to make LMs easily understand the chemically meaningful semantics. Moreover, the process of generating explanations should reflect the reasoning process behind the model architecture. Therefore, by disentangling the information flows of Transformer-based LMs, we integrate the self-attention weights and gradients to capture the substructure interactions to better quantify each chemically meaningful substructure's contribution to the predicted molecular properties.

2. To make the explanations more faithfully respect the structure-property relationships, we elaborate on one marginal loss to calibrate the explanations by aligning them with the chemists' annotations. We show that using only a few molecules with ground truth annotations can significantly improve the explanation accuracy by up to 5%.

3. We first bridge the manifold hypothesis with explainable molecular property prediction. We theoretically demonstrate that the elaborated marginal loss aligns explanations with the data manifold, respecting the structure-property relationship.

Experimental results over six mutagenicity datasets and one hepatotoxicity dataset demonstrate that *Lamole* can achieve comparable classification accuracy and improve the explanation accuracy by up to $14.3\%$. We also quantitatively evaluate explanations based on the first proposed plausibility metric. Compared to alternative baselines, the explanation plausibility of *Lamole* is improved by up to $9\%$. Extensive experimental studies demonstrate *Lamole* achieves state-of-the-art performance in explainable molecular property prediction.

## 2 RELATED WORK

Several explainable GNNs are proposed to explain the relationship between the input graph and the prediction (Sun et al., 2023; Xiong et al., 2019; Lin et al., 2022; Ying et al., 2019; Luo et al., 2020; Lin et al., 2021). Among these works, structure similarity or attention weights are proposed to capture structural interaction (Sun et al., 2023; Xiong et al., 2019). However, two similar substructures do not necessarily lead to interaction between them, and attention weights are often inconsistent with the feature importance (Jain & Wallace, 2019; Serrano & Smith, 2019; Abnar & Zuidema, 2020). In addition, as shown in Fig. 1 (b)∼(d), some trivial structures received relatively high importance scores, indicating the explanations might not align well with the chemical concepts.

On the other hand, with string-based molecular representations, LMs show great potential in molecular property prediction (Chithrananda et al., 2020; Wang et al., 2019; Ahmad et al., 2022; Ross et al., 2022). However, the "black-box" characteristics of LMs hamper trust use of these potent computational tools in scientific domains. Some explainability techniques could be applied to LMs. One way is to use the attention weights over the input tokens. However, recent studies suggest that "attention is not explanation" because attention weights could not reflect the true feature importance (Jain & Wallace, 2019; Serrano & Smith, 2019; Abnar & Zuidema, 2020). Perturbation-based methods perturb the inputs and evaluate the output changes to reveal the input importance. However, the generated explanations may change drastically with very small perturbations (Agarwal et al., 2021). Gradient-based methods determine the feature importance by the partial derivatives of the output to each feature (Selvaraju et al., 2017). However, several works show that the gradient-based methods may not be reliable, as they disregard the influence of model architectures on the output and fail to incorporate the information of the model architectures into the explanations (Adebayo et al., 2018; Agarwal et al., 2021; Rudin, 2019). Therefore, the explanation generation process should reflect the model reasoning process behind the model architectures. To this end, we disentangle the model architectures' information flows to generate explanations that faithfully reveal the structure-property relationship.

## 3 METHODOLOGY

**Problem Setup.** Given a dataset $\mathcal{G} = \{(g^{(i)}, y^{(i)})\}$ consisting of molecular graphs $\{g^{(i)}\}$ with their property labels $\{y^{(i)}\}$, explainable molecular property prediction aims to train a model $f$

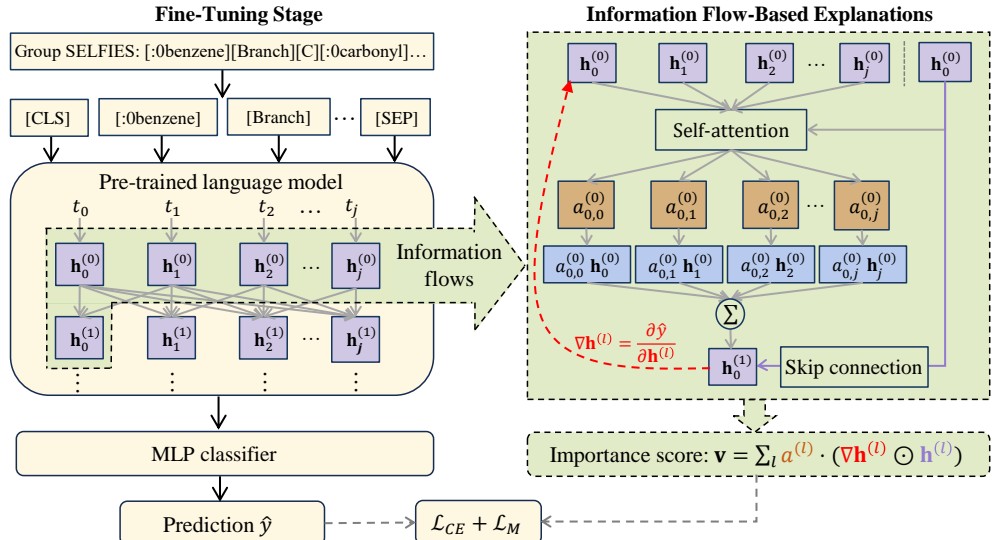

Figure 3: An illustration of *Lamole*. Left panel: Group SELFIES strings are tokenized for fine-tuning the pre-trained language model, and an MLP classifier is equipped with a cross-entropy loss $\mathcal{L}_{CE}$ for molecular property prediction. Right panel: We disentangle the information flows of the Transformer to assert that both attention weights and gradient determine the output. Therefore, we incorporate the attention weights and gradients together to generate importance scores $\mathbf{v}$ as explanations. In addition, a marginal loss $\mathcal{L}_M$ is designed to align explanations with the chemists' annotations $\mathbf{m}$.

to map a molecule $g$ to its property $y$, denoted as denoted as $f : g \mapsto y$, while providing an importance score vector $\mathbf{v}^{(i)} = \{\mathbf{v}_1^{(i)}, \ldots, \mathbf{v}_j^{(i)}\}$ to indicate the contribution of $j$-th functional group/fragment to its property $y$. Particularly, this work proposes to convert the molecular graph $g^{(i)}$ into the Group SELFIES string represented as $s^{(i)} = \{t_1^{(i)}, \ldots, t_j^{(i)}\}$, where $t_j^{(i)}$ is the $j$-th functional group/fragment's token, i.e., [·] in Fig. 2. In addition, this work calibrates the explanations in a supervised manner. For this purpose, a few molecules with annotation masks are provided. The annotation masks $\mathbf{m}(g^{(i)}) \in \{0, 1\}^j$ indicate whether $t_j^{(i)}$ is the token of a ground truth substructure, where $\mathbf{m}_j(g^{(i)}) = 1$ denotes the substructure corresponding to the token $t_j^{(i)}$ inducing the molecular property of $g^{(i)}$. This work uses $\mathcal{D} = \{(s^{(i)}, y^{(i)})\}$ with a few annotation masks to learn a model $f$ for explainable molecular property prediction. We omit the superscript $^{(i)}$ for simplicity in the following parts.

## 3.1 OUR DESIGN: *Lamole*

In this work, we pre-train Transformer-based LMs, e.g., Bert family models, using the Group SELFIES corpus to make the models understand the chemical semantics behind Group SELFIES strings. The details of the pre-train stage are shown in Appendix A.3 Then, we fine-tune the LMs with Group SELFIES strings and the molecular property labels for explainable molecular property prediction. An illustration of proposed *Lamole* is in Fig. 3. In what follows, we will introduce the detailed design of *Lamole*.

**Fine-Tuning Stage.** Fig. 3 shows the fine-tuning stage of the proposed *Lamole*. We assume that the Transformer encoder in *Lamole* stacks $L$ identical Transformer layers to encode the molecular string $s$ as token embedding $\mathbf{h}^{(l)} = \{\mathbf{h}_1^{(l)}, \ldots, \mathbf{h}_j^{(l)}\}$, where $\mathbf{h}^{(l)}$ is the token embedding at the $l$-th layer. We use the self-attention weighted average embedding $\mathbf{h}_o = (\sum_{j=1} \alpha_j \cdot \mathbf{h}_j^{(L)})/(\sum_{j=1} \alpha_j)$ for molecular property prediction, where $\alpha_j$ is the attention weight of the $j$-th token. A multilayer perceptron (MLP) classifier is added to predict the molecular property $\hat{y} = \text{MLP}(\mathbf{h}_o)$ by minimizing the classification cross-entropy loss $\mathcal{L}_{CE}(y, \hat{y})$.

**Information Flow-Based Explanations.** To derive the explanations from Transformer-based LMs, a common practice is to use gradient-based methods to determine each feature's importance by analyzing the output's partial derivatives to each input feature (Selvaraju et al., 2017). However, we argue that using gradients alone cannot effectively capture substructure interactions. As depicted in Fig. 1 (f), the gradient-based method CradCAM incorrectly attributes the property to the nitro group and a benzene ring that is not connected to the nitro group.

We illustrate the possible reason by disentangling the information flows of the Transformer. As shown in Fig. 3 right panel, due to the skip connection and attention mechanism, both attention weights, gradients, and the input contribute to the outputs. Therefore, using gradients alone as explanations could fall short of capturing interactions. To address the issue, we leverage both attention and gradients, as well as the input, to derive explanations. Below, we will elaborate on integrating attention weights into gradient-based explanations.

Firstly, we show the process of deriving the gradient-based explanations. Similar to GradCAM (Selvaraju et al., 2017), the gradient with respect to the $j$-th token's embedding $\mathbf{h}_j^{(l)}$ at the $l$-th layer is derived by $\nabla \mathbf{h}_j^{(l)} = \partial \hat{y} / \partial \mathbf{h}_j^{(l)}$, where $\nabla \mathbf{h}_j$ signifies the importance of the $j$-th token in relation to the predicted property $\hat{y}$. Due to the skip connection in Fig. 3 right panel, the input, and its corresponding gradient should be leveraged together, and the weighted importance $\mathbf{w}$ of the $j$-th token at the $l$-th layer can be determined by

$$\mathbf{w}_j^{(l)} = \nabla \mathbf{h}_j^{(l)} \odot \mathbf{h}_j^{(l)}, \tag{1}$$

where $\odot$ is the Hadamard product. The weighted importance is regarded as the gradient-based explanation.

The interaction among tokens can be revealed by the self-attention mechanism in Fig. 3 right panel. The attention mechanism calculates pairwise similarity scores between all pairs of tokens to determine attention weights, and these attention weights inherently encode the functional group interactions. Therefore, we combine the attention weights with the gradient-based explanation to capture functional group interactions. Assuming the attention weights of the $j$-th token at the $l$-th layer is $\alpha_j^{(l)}$, we integrate the attention weights with weighted importance $\mathbf{w}$ to consider the interactions. The importance score of the molecule $g$ can be obtained by

$$\mathbf{v}_j^{(l)}(g) = \left( \tanh(\overline{\alpha_j^{(l)}}) \cdot \tanh(\mathbf{w}_j^{(l)}) \right)^{\frac{1}{2}}, \tag{2}$$

where $\overline{\alpha_j^{(l)}}$ is the averaging of attention weights of multiple attention heads. Finally, we sum $\mathbf{v}_i^{(l)}$ over all layers as the final importance score of the $j$-th token,

$$\mathbf{v}(g) = \text{softmax}(\sum_{l=1}^{L} \mathbf{v}^{(l)}(g)). \tag{3}$$

The higher the importance scores, the greater the contribution of the corresponding functional groups/fragments to the molecular property.

**Towards Plausible Explanations.** One plausible explanation should faithfully uncover the structure-property relationships. In other words, the explanation should match the ground-truth substructures with high confidence. Nevertheless, the importance scores of ground-truth substructures might not be significantly higher than those of other parts. To address this issue, we propose a marginal loss to explicitly align explanations with the chemists' annotations to improve the explanations' plausibility.

First, we formally define the plausibility of explanations. "Plausibility" refers to how the interpretation convinces humans (Wiegreffe & Pinter, 2019; Herman, 2017; Jacovi & Goldberg, 2020). Similarly, in our context, "plausibility" refers to the degree of confidence in the explanations that would convince the chemists.

**Definition 1** *(Plausibility): Given the importance scores $\mathbf{v}$ over all tokens in the molecule $g$, the mean importance score $\mathbf{v}_{\in \mathcal{T}_g}$ over ground truth substructures $\mathcal{T}_g$ and the mean importance score $\mathbf{v}_{\notin \mathcal{T}_g}$ over other substructures $\mathcal{T}_g$ are denoted by $\mathbf{v}_{\in \mathcal{T}_g} = \frac{\sum_j \mathbf{v}_j \cdot \mathbb{I}(t_j \in \mathcal{T}_g)}{\sum_j \mathbb{I}(t_j \in \mathcal{T}_g)}$ and $\mathbf{v}_{\notin \mathcal{T}_g} = \frac{\sum_j \mathbf{v}_j \cdot \mathbb{I}(t_j \notin \mathcal{T}_g)}{\sum_j \mathbb{I}(t_j \notin \mathcal{T}_g)}$, respectively, where $\mathbb{I}(\cdot)$ is the indicator function. The explanations' plausibility $\mathrm{EP}(g)$ is defined as the ratio of the difference between $\mathbf{v}_{\in \mathcal{T}_g}$ and $\mathbf{v}_{\notin \mathcal{T}_g}$ to $\mathbf{v}_{\notin \mathcal{T}_g}$,*

$$\mathrm{EP}(g) = \frac{\mathbf{v}_{\in \mathcal{T}_g} - \mathbf{v}_{\notin \mathcal{T}_g}}{\mathbf{v}_{\notin \mathcal{T}_g}}. \tag{4}$$

The higher the EP value, the greater the confidence of the explanation in matching the ground truth substructure.

Eq. (4) defines the explanation plausibility based on the scores of two parts, i.e., the scores on ground truth and the scores on non-ground truth. The lower the scores on non-ground truth and the greater the scores on ground truth, the better explanation plausibility. Therefore, to maximize the plausibility, our objective can be transformed to minimize the importance score of non-ground truth and maximize the importance score of ground truth.

To this end, we design a max-margin loss to optimize the importance score. In our work, the ground truth substructures are annotated by a binary mask vector $\mathbf{m}(g) \in \{0, 1\}^j$. It is worth noting that using only a few annotations can significantly improve the explanation accuracy. Specifically, the mask vector $\mathbf{m}$ enforces the explanations to align with the ground truth substructures. To achieve the goal, a max-margin loss is designed by maximizing the mean value of the importance scores of tokens that have mask values of 1 while minimizing the mean value of importance scores for tokens with mask values of 0.

$$\mathcal{L}_M(\mathbf{v}, \mathbf{m}) = \mathbb{E}_{g \in \mathcal{G}} \left[ \max \left( 0, \frac{\sum_{j=1}(1 - \mathbf{m}_j(g)) \cdot \mathbf{v}_j(g)}{N_s} - \frac{\sum_{j=1} \mathbf{m}_j(g) \cdot \mathbf{v}_j(g)}{N_c} \right) + \triangle_1 \right], \tag{5}$$

where $\triangle_1$ is a margin term, $N_s$ is the number of tokens with mask values $\mathbf{m}(g)$ of 0, and $N_c$ is the number of tokens with mask values $\mathbf{m}(g)$ of 1. The overall optimization objective of the fine-tuning stage is to minimize $\mathcal{L}_{CE} + \mathcal{L}_M$. The core of Eq. (5) is the discrepancy between the average importance score of ground truth and the average importance score of non-ground truth. By minimizing $\mathcal{L}_M$, the discrepancy between the two importance scores is maximized. In other words, the average importance score of non-ground truth is suppressed, and the average importance score of ground truth is increased. Finally, the explanation plausibility defined in Eq. (4) is improved. The next section will theoretically show that by using the designed marginal loss, the explanations can faithfully reflect the structure-property relationships.

## 3.2 THEORETICAL ANALYSIS

We bridge the manifold hypothesis with the marginal loss to theoretically show that the explanations can respect the structure-property relationships. Before giving the proof, the notation and definition regarding the manifold hypothesis are presented.

**Manifold Hypothesis.** It is widely believed that natural data, including molecules, distribute around a manifold (Bordt et al., 2023; Lin et al., 2022; Godwin et al., 2022; Singh et al., 2020). According to the manifold hypothesis for gradient-based explanations (Bordt et al., 2023), if a feature lies in the tangent space of a manifold, then the feature respects the manifold and contributes to the class, and such a feature is desirable to be explained. We call these features "causal features" in our work. Conversely, if a feature is orthogonal to the manifold, then the feature does not contribute to the class. We call these features "spurious features".

With the annotation masks, the causal features $s^*$ and spurious features $\overline{s}^*$ can be distinguished by $s^* = s \odot \mathbf{m}(g)$ and $\overline{s}^* = s \odot (1 - \mathbf{m}(g))$, respectively, where $\overline{s}^* \cup s^* = s$ and $s^* \cap \overline{s} = \varnothing$. By projecting the causal features and spurious features into the data manifold $\mathcal{M}$, the corresponding manifold regarding the causal features and spurious features can be defined as follows,

**Definition 2** *(Causal feature manifold and spurious feature manifold): Assume the distribution $p(g|y)$ is implicitly modeled by a manifold $\mathcal{M}$, and the manifold can be decomposed into two components,*

$$\underbrace{p(g|y)}_{\mathcal{M}} = \underbrace{p(g|y) \odot \mathbf{m}(g)}_{\mathcal{M}_c} + \underbrace{p(g|y) \odot (1 - \mathbf{m}(g))}_{\mathcal{M}_s}, \tag{6}$$

*where $\mathcal{M}_c$ is the causal feature manifold and $\mathcal{M}_s$ represents the spurious feature manifold.*

With this decomposition, we demonstrate how the gradient-based explanations $\nabla_g \log p(y|g)$ can uncover the structure-property relationships. Due to the page limitation, we provide the proof in Appendix A.1.

**Theorem 1** *The marginal loss of Eq. (5) aligns the gradient-based explanations $\nabla_g \log p(y|g)$ with the tangent space of the causal feature manifold $\mathcal{M}_c$, thus respecting the structure-property relationships.*

## 4 EXPERIMENTS

### 4.1 EXPERIMENTAL SETUP

**Datasets**. We use six datasets on two types of tasks, i.e., hepatotoxicity and mutagenicity, to evaluate the algorithmic performance for the explainable molecular property prediction. Six mutagenicity datasets are Mutag (Debnath et al., 1991), Mutagen (Morris et al., 2020), PTC-FM (Toivonen et al., 2003), PTC-FR (Toivonen et al., 2003), PTC-MM (Toivonen et al., 2003), and PTC-MR (Toivonen et al., 2003). For hepatotoxicity (Toivonen et al., 2003), the Liver dataset (Liu et al., 2015) is used. We used these datasets as the ground truth substructures in these datasets are known. For detailed information on these datasets and the ground truth substructures, please refer to Appendix A.2.

**Evaluation Metrics.** We use three metrics for evaluating the performance of the proposed *Lamole*. *1) Classification Accuracy*: We evaluate the model's predictions by $\sum_{i=1}^{I} \mathbb{I}(y^{(i)} = \hat{y}^{(i)})/I$. *2) Explanation Accuracy*: We follow the experimental settings in GNNExplainer (Ying et al., 2019), which formulates the explanation problem as a binary classification of edges. We treat edges inside ground-truth substructure as positive edges and negative otherwise, and AUC is adopted as the metric for quantitative evaluation. We only consider the mutagenic/hepatotoxic molecules because no explicit substructures exist in nonmutagenic/nonhepatotoxic ones. *3) Explanations' Plausibility*: We use the defined explanations' plausibility EP to measure how confident the explanation aligns with the ground truth.

**Baselines.** We combine *Lamole* into three BERT family models, such as DistilBert (Sanh, 2019), DeBerta (He et al., 2020), and Bert to evaluate the performance of the proposed *Lamole*. For evaluating classification accuracy, we compare our *Lamole* with one SMILES string-based LM, ChemBERTa (Chithrananda et al., 2020) and several GNNs including GCN (Kipf & Welling, 2016), DGCNN (Zhang et al., 2018), edGNN (Jaume et al., 2019), GIN (Xu et al., 2018), RW-GNN (Niko-lentzos & Vazirgiannis, 2020), DropGNN (Papp et al., 2021), and IEGN (Maron et al., 2018).

For evaluating explanation accuracy, *Lamole* is compared with three types of alternative methods: 1) GCN with feature-based explainability techniques, including SmoothGrad (Smilkov et al., 2017), GradInput (Shrikumar et al., 2017), and GradCAM (Selvaraju et al., 2017), 2) Bert with the above feature-based explainability techniques, where Group SELFIES is used as input for a fair comparison, and 3) explainable GNNs including OrphicX (Lin et al., 2022), GNNExplainer (Ying et al., 2019), PGExplainer (Luo et al., 2020), and Gem (Lin et al., 2021). The details of experimental settings can be found in Appendix A.3.

### 4.2 RESULTS

**Prediction Performance**. Table 1 shows the classification accuracy of compared algorithms. As we can see, our proposed *Lamole*+DistilBert, *Lamole*+DeBerta, and *Lamole*+Bert not only can provide explainability but also can achieve comparable prediction accuracy as compared to existing predictive methods. In addition, *Lamole* models show superior performance over ChemBERTa. This suggests

Table 1: Mean Classification Accuracy on the Seven Datasets (%)

| Methods | Mutag | Mutagen | PTC-FM | PTC-FR | PTC-MM | PTC-MR | Liver |
|---|---|---|---|---|---|---|---|
| GCN (Kipf & Welling, 2016) | 84.6 | 78.9 | 54.8 | 63.0 | 57.8 | 53.3 | 41.1 |
| DGCNN (Zhang et al., 2018) | 85.8 | 74.8 | 57.3 | 63.5 | 61.0 | 58.6 | 44.6 |
| edGNN (Jaume et al., 2019) | 86.9 | 75.2 | 59.8 | 65.7 | $64.4^\dagger$ | 56.3 | 44.5 |
| GIN (Xu et al., 2018) | 87.5 | $82.3^\ddagger$ | $62.1^\dagger$ | 66.2 | $65.1^\ddagger$ | 64.0 | 44.9 |
| RW-GNN (Nikolentzos & Vazirgiannis, 2020) | 87.2 | 80.3 | 61.9 | 64.0 | 62.4 | 57.0 | 43.2 |
| DropGNN (Papp et al., 2021) | $89.4^\ddagger$ | $80.7^\dagger$ | 62.0 | 66.0 | 63.7 | 64.2 | 45.0 |
| IEGN (Maron et al., 2018) | 84.6 | 80.1 | 60.8 | 59.8 | 61.1 | 59.5 | 45.3 |
| ChemBERTa (Chithrananda et al., 2020) | 86.8 | 78.0 | 60.0 | 65.7 | 60.4 | 58.7 | 45.7 |
| *Lamole*+DistilBert | 84.2 | 76.8 | 57.5 | 69.0 | 60.2 | $64.5^\dagger$ | $47.2^\dagger$ |
| *Lamole*+DeBerta | 86.8 | 73.7 | 58.6 | $69.5^\dagger$ | 59.7 | 63.8 | 45.8 |
| *Lamole*+Bert | $88.2^\dagger$ | 74.5 | $62.4^\ddagger$ | $70.0^\ddagger$ | 61.2 | $66.0^\ddagger$ | $47.5^\ddagger$ |

$\ddagger$ and $\dagger$ denote the best and the second-best results, respectively.

that using molecular representations with more chemical semantics, like Group SELFIES, can help LMs better learn the chemical semantics and structure-property relationships.

**Explanation Performance.** Table 2 presents the explanation accuracy of the compared explainability techniques. It should be noted that the ground truth annotations used in our work provide additional supervisory signals. Therefore, we also align the generated explanations of these baselines with the annotations for fair comparison, and the ground truth annotation rate is 10%. *Lamole* improves the explanation accuracy by $1.4\% \sim 14.3\%$ compared to the baseline methods. We also investigated the explanation accuracy of our *Lamole* under different ground truth annotation rates (10%, 20%, 50%, and 100%). The impact of annotation rates can be found in Appendix A.5. In addition, we discuss the rationale of using labeled annotations (see Appendix A.4).

Table 2: Mean Explanation Accuracy on the Seven Datasets (%)

| Methods | Mutag | Mutagen | PTC-FM | PTC-FR | PTC-MM | PTC-MR | Liver |
|---|---|---|---|---|---|---|---|
| GradInput+GCN (Shrikumar et al., 2017) | 70.3 | 67.9 | 69.7 | 66.4 | 64.6 | 65.0 | 73.0 |
| GradCAM+GCN (Selvaraju et al., 2017) | 69.8 | 67.0 | 71.0 | 67.9 | 66.2 | 67.3 | 69.4 |
| SmoothGrad+GCN (Smilkov et al., 2017) | 69.2 | 66.8 | 67.5 | 62.6 | 64.9 | 63.1 | 66.4 |
| GradInput+Bert (Shrikumar et al., 2017) | 75.1 | 72.6 | 73.0 | 68.9 | 65.6 | 69.6 | 75.6 |
| GradCAM+Bert (Selvaraju et al., 2017) | 75.3 | 72.4 | 77.5 | 70.0 | 70.2 | 73.0 | 76.0 |
| SmoothGrad+Bert (Smilkov et al., 2017) | 73.4 | 72.8 | 73.7 | 71.0 | 67.0 | 69.9 | 75.1 |
| GNNExplainer (Ying et al., 2019) | 70.6 | 64.2 | 68.9 | 67.9 | 66.8 | 67.1 | 72.1 |
| PGExplainer (Luo et al., 2020) | 66.5 | 58.7 | 70.3 | 68.0 | 65.9 | 67.0 | 71.5 |
| Gem (Lin et al., 2021) | 73.7 | 66.0 | 71.3 | 69.0 | 68.9 | 69.2 | 73.6 |
| OrphicX (Lin et al., 2022) | $78.0^\ddagger$ | 71.5 | 74.6 | 70.4 | $70.9^\dagger$ | 71.4 | 74.0 |
| *Lamole*+DistilBert | 70.9 | 73.0 | 74.0 | 70.2 | 69.6 | $78.1^\ddagger$ | $76.1^\ddagger$ |
| *Lamole*+DeBerta | 76.1 | $75.0^\dagger$ | $79.9^\dagger$ | $72.1^\dagger$ | 70.3 | $77.2^\dagger$ | 75.0 |
| *Lamole*+Bert | $77.8^\dagger$ | $75.2^\ddagger$ | $81.1^\ddagger$ | $72.2^\ddagger$ | $72.0^\ddagger$ | 73.1 | $77.3^\dagger$ |

$\ddagger$ and $\dagger$ denote the best and the second-best results, respectively.

We selected some representative molecules for explanation visualization. These explanations are shown in Figs. 1, 4, 12, 13, 14, and 15, respectively. The right panel of those figures is the importance scores obtained by *Lamole*, where "other" in the figures is the average importance score of other unlisted functional groups/fragments. Compared to baseline methods, *Lamole* provides chemically meaningful explanations. Particularly, the interaction among the functional groups is successfully captured. More visualization results can be found in Appendix A.7.

In addition, we evaluated the performance of compared algorithms by using the proposed explanation plausibility metric EP. The statistical results of EP are presented in Figs. 5 and 11. From the figures, we can observe that the EP values of the comparison algorithm are slightly lower, which means that the algorithms cannot confidently reflect the relationships between structure and property. Compared to the comparison algorithm, the EP values of *Lamole* have increased by $2\% \sim 9\%$. More analysis regarding the explanation plausibility can be found in Appendix A.6.

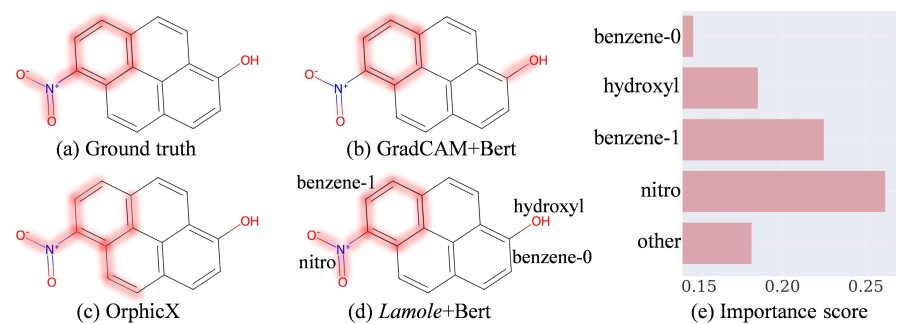

Figure 4: Explanation visualization of one molecule (ID: 155) from the Mutag dataset.

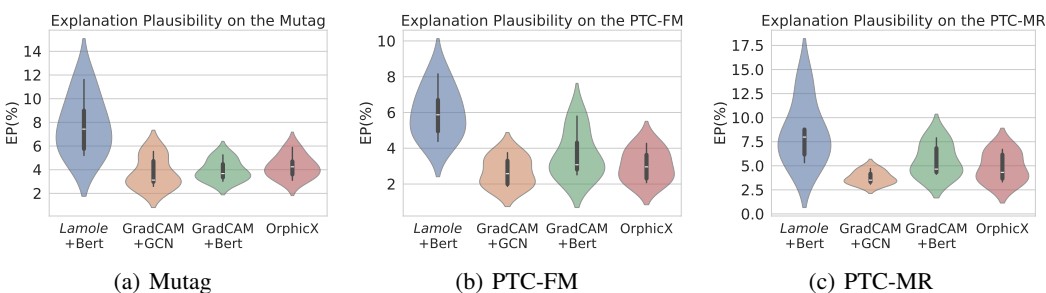

Figure 5: The explanation plausibility of the compared algorithms on the Mutag, PTC-FM, and PTC-MR datasets.

### 4.3 ABLATION STUDIES

We conducted ablation studies for each component in our *Lamole*. Specifically, we removed the attention weights in the explanations, removed the marginal loss, and only used attention weights as explanations. The corresponding ablation algorithms are named *Lamole* ($-\alpha$), *Lamole* ($-\mathcal{L}_M$), and *Lamole* (Att), respectively. The results are shown in Fig. 6. The explanation accuracy decreases by 1.4%~2.3% when removing the attention weights. Removing the marginal loss can decrease the explanation accuracy by 1.0%~5.0%. Regarding the results of using only attention weights, The explanation accuracy decreases by 1.4%~6.3%. The above results confirm the effectiveness of using the marginal loss, attention weights, and gradients.

From the perspective of model training, the marginal loss enables the model to be trained under the causal signals. To verify, we compared the classification performance with and without the

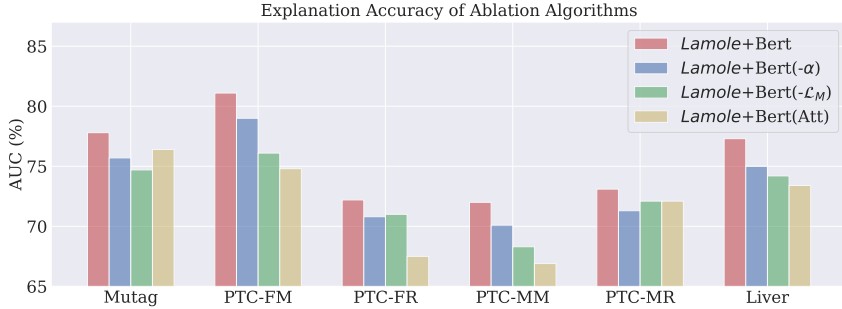

Figure 6: The explanation accuracy of *Lamole*+Bert, *Lamole*+Bert ($-\alpha$), *Lamole*+Bert ($-\mathcal{L}_M$), and *Lamole* (Att).

marginal loss, as shown in Table 3. Without the marginal loss, the classification accuracy degrades by 0.7%~3.7%. The above results indicate that marginal loss could help identify the causal features, thereby improving classification accuracy.

Table 3: The classification performance with and without the marginal loss (%)

| Methods | Mutag | Mutagen | PTC-FM | PTC-FR | PTC-MM | PTC-MR | Liver |
|---------|-------|---------|--------|--------|--------|--------|-------|
| *Lamole-$\mathcal{L}_M$* | 84.9 | 73.8 | 58.6 | 69.3 | 59.7 | 62.3 | 44.6 |
| *Lamole* | **88.2** | **74.5** | **62.4** | **70.0** | **61.2** | **66.0** | **47.5** |

To investigate the attention weights, the attention weights of two molecules are depicted in Fig. 7. From Fig. 7 middle panel, it can be found the correlation among the ground truth substructures is higher than others, showcasing the rationality of using attention weights to capture the functional group interactions. When we aggregate the attention weights for each token, the top two attention weights of the molecule (ID:155) match the two ground truth substructures (see Fig. 7 (a) right panel). However, the top two attention weights of the molecule (ID:156) do not match the two ground truth substructures (see Fig. 7 (b) right panel). The above results indicate that attention weights can capture the interactions and also confirm that "attention is not explanation" (Jain & Wallace, 2019; Serrano & Smith, 2019; Abnar & Zuidema, 2020). Due to the page limitation, the limitation of the proposed work is provided in Appendix A.9.

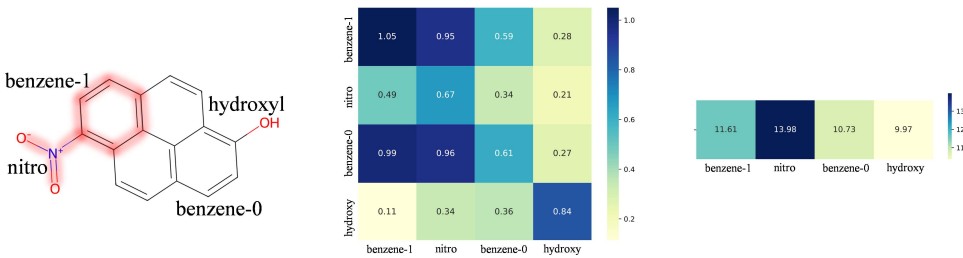

(a) A molecule (ID: 155) from the Mutag dataset

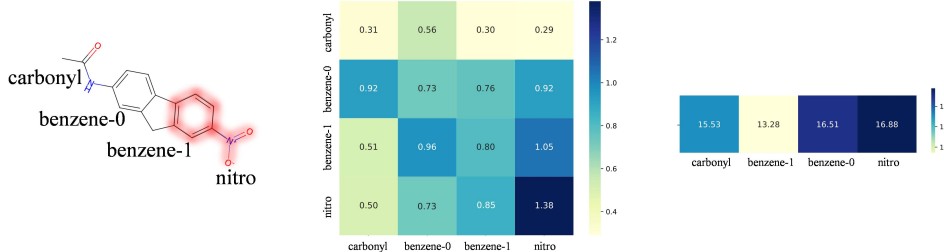

(b) A molecule (ID: 156) from the Mutag dataset

Figure 7: Attention weights visualization. The ground truth substructures are highlighted in red.

## 5 CONCLUSIONS

This work proposed *Lamole* for explainable molecular property prediction based on language models. *Lamole* uses Group SELFIES as input for chemically meaningful semantics. By disentangling the information flows of Transformer-based LMs, *Lamole* integrates attention weights into gradients to generate explanations to quantify each chemically meaningful substructure's impact on the model's output. Furthermore, one marginal loss is designed to calibrate the explanations to be more faithful by aligning them with the chemists' annotation. *Lamole* 's effectiveness has been demonstrated through theoretical analysis and extensive experimental validation.

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

# A   APPENDIX

## CONTENTS

## A.1   BRIDGING MANIFOLD HYPOTHESIS WITH CHEMICAL CONCEPTS-ALIGNED EXPLANATIONS

**Proof of Theorem 1**: The gradient with respect to the prediction $\nabla_g \log p(y|g)$ can be decomposed into the gradient on the causal features and spurious features, respectively,

$$\nabla_g \log p(y|g) = \nabla_g \log p(y|g) \odot \mathbf{m}(g) + \nabla_g \log p(y|g) \odot (1 - \mathbf{m}(g)). \tag{7}$$

By minimizing the loss of Eq. (5), the gradients on spurious features $\nabla_g \log p(y|g) \odot (1 - \mathbf{m}(g))$ are suppressed, and $\nabla_g \log p(y|g)$ approximates $\nabla_g \log p(y|g) \odot \mathbf{m}(g)$. Therefore, we have $\nabla_g \log p(y|g) \approx \nabla_g \log p(y|g) \odot \mathbf{m}(g)$. On the other hand, $\nabla_g \log p(y|g) \odot \mathbf{m}(g)$ can be rewritten as

$$\nabla_g \log p(y|g) \odot \mathbf{m}(g) = \nabla_g \log p(g|y) \odot \mathbf{m}(g) - \sum_j p(y = j|g) \nabla_g \log p(g|y = j) \odot \mathbf{m}(g). \tag{8}$$

Because the data distribution $p(g|y) \odot \mathbf{m}(g)$ reflects the causal feature manifold $\mathcal{M}_c$, the gradient of the distribution $\nabla_g p(g|y) \odot \mathbf{m}(g)$ represents the tangent space of the causal feature manifold $\mathcal{M}_c$. In addition, Eq. (8) shows that the $\nabla_g \log p(y|g) \odot \mathbf{m}(g)$ is a linear combination of $\nabla_g p(g|y) \odot \mathbf{m}(g)$, so $\nabla_g \log p(y|g) \odot \mathbf{m}(g)$ also lies tangent space of the manifold $\mathcal{M}_c$.

Together with Eq. (7) and Eq. (8), we prove the gradient-based explanations $\nabla_g \log p(y|g)$ lies tangent space of the manifold $\mathcal{M}_c$. This indicates by minimizing the loss of Eq. (5), the model $p(y|g)$ has reflected the causal feature manifold. According to the manifold hypothesis, the features on the causal feature manifold contribute to the molecular property. Therefore, the gradient-based explanations $\nabla_g \log p(y|g)$ can uncover the causal features, thus revealing the structure-property relationships. This completes the proof.

## A.2   DATASETS

We use six datasets on two types of tasks, i.e., hepatotoxicity and mutagenicity, to evaluate the algorithmic performance for the explainable molecular property prediction. Six mutagenicity datasets are Mutag (Debnath et al., 1991), Mutagen (Morris et al., 2020), PTC-FM (Toivonen et al., 2003), PTC-FR (Toivonen et al., 2003), PTC-MM (Toivonen et al., 2003), and PTC-MR (Toivonen et al., 2003). For hepatotoxicity (Toivonen et al., 2003), the Liver dataset (Liu et al., 2015) is used. Larger-sized molecules typically include more complex structures. The datasets that we used contained relatively large molecules. The maximal number of atoms of Mutag, Mutagen, PTC-FM, PTC-FR, PTC-MM, PTC-MR, and Liver are 26, 417, 64, 64, 64, 64, and 157, respectively. The details of the used dataset are provided in Table 4.

Table 4: Statistiscal Information of the Datasets

| Datasets | Mutag | Mutagen | PTC-FM | PTC-FR | PTC-MM | PTC-MR | Liver |
|---|---|---|---|---|---|---|---|
| Graphs | 188 | 4337 | 349 | 351 | 336 | 344 | 587 |
| Classes | 2 | 2 | 2 | 2 | 2 | 2 | 3 |
| Max nodes | 26 | 417 | 64 | 64 | 64 | 64 | 157 |
| Avg nodes | 17.9 | 29 | 14.1 | 14.6 | 14 | 14.3 | 25.6 |
| Avg edges | 19.8 | 30 | 14.5 | 15 | 14.3 | 14.7 | 27.4 |
| Ground truth* | 120 | 724 | 58 | 49 | 51 | 61 | 187 |

* denotes the number of molecules with known ground truth substructures.

Following OrphicX (Lin et al., 2022), on the Mutag, Mutagen, PTC-FM, PTC-FR, PTC-MM, and PTC-MR datasets, we only consider the explanations for the mutagenic class, because the molecules of the non-mutagenic class have no ground truth. Although some works used single $N = N$, $NO_2$, or $NH_2$ as ground truth, this is not reasonable, as 32% of non-mutagenic graphs in Mutagen containing at least single $NO_2$ or $NH_2$. In fact, the ground truth for the mutagenic class is the benzene with a chemical group on it, such as $N = N$, $NO_2$, and $NH_2$ (Lin et al., 2021; 2022; Patlewicz et al., 2003).

For the Liver dataset, the molecules of possible hepatotoxicity with ground truth substructures and hepatotoxicity with ground truth substructures are collected for explainable molecular property prediction. The twelve ground truth substructures of the Liver dataset are shown in Fig. 8.

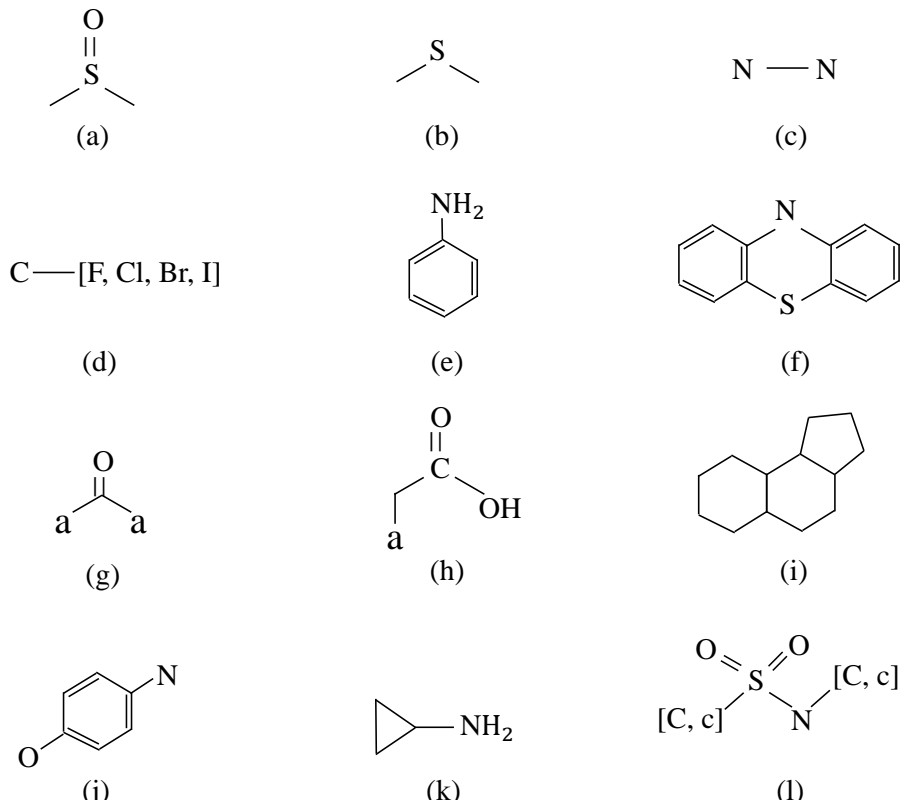

Figure 8: Twelve ground truth substructures of the Liver dataset. Lowercase element symbols represent aromatic atoms of the element; the letter "a" matches any aromatic atom. Elements in square brackets match any of the elements in a molecule.

Users can specify the fragments in Group SELFIES they want to cover by using a dictionary. The dictionary is called "group set". Our group set covered several fundamental functional groups, including benzene, amido, carboxyl, hydroxyl, nitro, amino, toluene, nitroso, cyan, and methyl.

### A.3 EXPERIMENTAL SETTINGS

Following ChemBERTa (Chithrananda et al., 2020), our BERT family models were pre-trained on a set of 100,000 molecules from the ZINC dataset (Irwin et al., 2012). During pre-training, 15% of tokens in each input string were randomly masked for masked language learning. For each dataset, the ratio of samples with annotation masks over the data size is 10%. The pre-training process was conducted for 10 epochs.

We finetuned the models on the Mutag, Mutagen, PTC-FM, PTC-FR, PTC-MM, PTC-MR, and Liver datasets for the downstream explainable molecular property prediction tasks. During the fine-tuning stage, for Mutag, Mutagen, and the four PTC datasets, we used an Adam optimizer with a learning rate of 5e-5 and weight decay of 1e-5. The number of epochs is set to 60. For the liver dataset, we used an Adam optimizer with a learning rate of 5e-7, and the other parameter settings were the same as the above. The margin term $\triangle_1$ in Eq. (5) is set to 1.

ChemBERTa is a SMILES string-based Bert model, and the input for ChemBERTa is SMILES strings. For SmoothGrad+Bert, GradInput+Bert, and GradCAM+Bert, the inputs are Group SELFIES strings for a fair comparison. For explainable GNNs and GCNs with feature-based explainability techniques, we select edges with the top-$K$ importance scores as the explanations, where $K$ is the number of edges in the corresponding ground truth substructures. For Bert with feature-based explainability techniques, we select tokens with the top-$K$ importance scores as the explanations, where $K$ is 2 for the Mutag, Mutagen, PTC-FM, PTC-FR, PTC-MM, PTC-MR, and Liver datasets, as 2 fragments (benzene with chemical groups such as $N = N$, $NO_2$, and $NH_2$ on the benzene) determine the mutagenic class. For the Liver dataset, the $K$ is the number of tokens of ground truth substructures. We conducted experiments on the computer with an NVIDIA A100 GPU.

### A.4 DISCUSSION ON USING LABELED ANNOTATIONS

*Lamole* requires human-labeled annotations. Due to the huge knowledge base of LLMs, we explored the use of LLMs, including ChatGPT and ChemLLM (Zhang et al., 2024), to annotate the ground truth. We input molecules' SMILES strings into the two LLMs to ask the ground truth. The explanation accuracy results are shown in Table 5.

It is obvious that there is a significant decrease in explanation accuracy, indicating that existing LLMs may make incorrect annotations. Therefore, we argue that this human-in-the-loop strategy — providing slight human annotations — to guide learning is reasonable and necessary for critical scientific domains.

Table 5: The explanation accuracy results when using different annotation methods

| Methods | Mutag | PTC-FM | PTC-FR | PTC-MM | PTC-MR | Liver |
|---|---|---|---|---|---|---|
| Lamole *(ChatGPT)* | 67.6 | 64.5 | 55.0 | 51.0 | 65.6 | 72.7 |
| Lamole *(ChemLLM)* | 62.5 | 72.1 | 61.5 | 57.1 | 71.9 | 72.7 |
| Lamole *(Human)* | **77.8** | **81.1** | **72.2** | **72.0** | **73.1** | **77.3** |

### A.5 MORE RESULTS ON DIFFERENT ANNOTATION RATES

The results of explanation accuracy of *Lamole* under different annotation rates (10%, 20%, 50%, and 100%) on the PTC-FR and PTC-MM datasets are shown in Fig. 9 and Fig. 10. It is clear that more annotations can constantly enhance the accuracy of the explanation. Compared to *Lamole* $(-\mathcal{L}_M)$, only using 10% molecules with ground truth annotations can significantly improve explanation accuracy by up to 5%. Using more annotations (from 10% to 20%) can achieve significant improvement in explanation accuracy. However, raising the rate from 50% to 100% can bring a limited increase in explanation accuracy on the four PTC datasets. Therefore, there is a trade-off between the explanation accuracy and additional annotation costs.

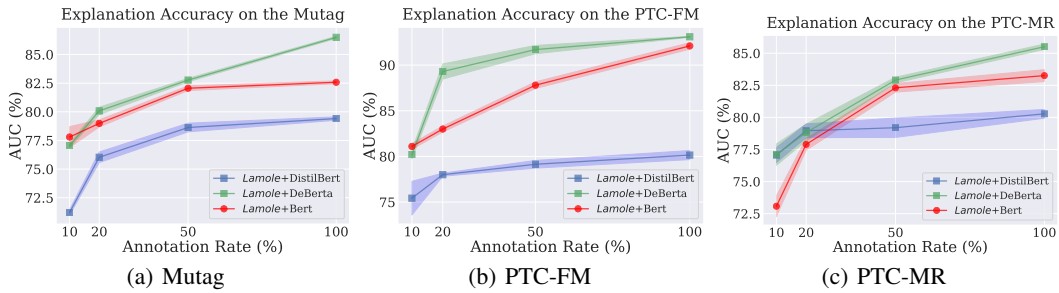

Figure 9: The explanation accuracy of *Lamole* with different annotation rates on the Mutag, PTC-FM, and PTC-MR datasets.

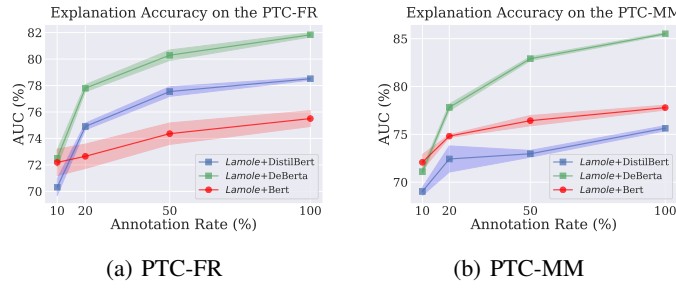

Figure 10: The explanation accuracy of *Lamole* with different annotation rates on the PTC-FR and PTC-MM datasets.

## A.6 MORE RESULTS ON EXPLANATION PLAUSIBILITY

The experimental results of the explanation plausibility on the PTC-FR, PTC-MM, and Liver datasets are presented in Fig. 5 and Fig. 11. The explanations' plausibility $\text{EP}(g)$ is defined as the ratio of the difference between the mean importance scores of ground truth and the mean importance scores of non-ground truth to the mean importance scores of non-ground truth. A larger ratio indicates that ground truth's importance scores exceed non-ground truth's. The high $\text{EP}(g)$ values of *Lamole* indicate *Lamole* can improve the importance scores of ground truth and suppress the importance scores of non-ground truth, leading to higher confidence in matching the ground truth.

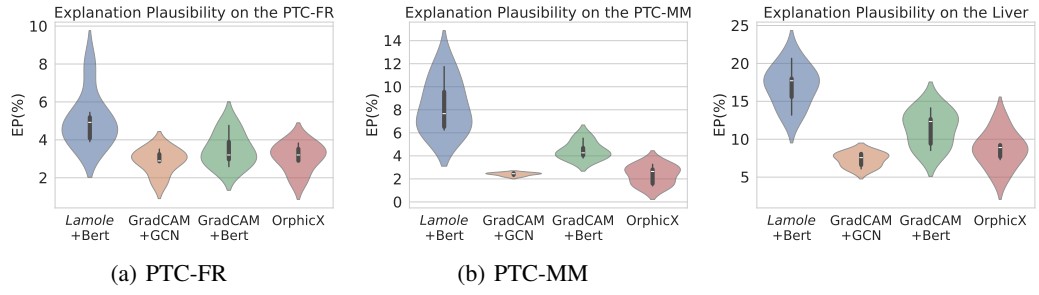

Figure 11: The explanation accuracy of *Lamole* with different annotation rates on the PTC-FR, PTC-MM, and Liver datasets.

## A.7    MORE VISUALIZATIONS

In addition to Figs. 1 and 4, the explanations of more molecules are shown in Figs. 12, 13, 14, and 15, respectively. The right panel of those figures displays the importance scores across the functional groups/fragments obtained by *Lamole*, where "other" represents the average importance score of the other unlisted functional groups/fragments. As shown in Figs. 13 and 14, *Lamole* accurately and confidently identify benzene with amido group and benzene-1 with nitro-1 group as explanations, respectively. While other methods can neither provide chemically meaningful explanations nor reflect the functional group interactions. These explanations demonstrate *Lamole*'s superior interpretation in faithfully revealing the structure-property relationships. However, as depicted in Fig. 15, although the ground truth substructures, i.e., benzene-1 and amido ground, are identified, other functional groups/fragments such as carbonyl-0, Br-0, and Br-1 also have relatively higher importance scores. This may be due to complex interactions caused by multiple functional groups. In future work, more strategies may need to be designed to reveal such complex functional group interactions.

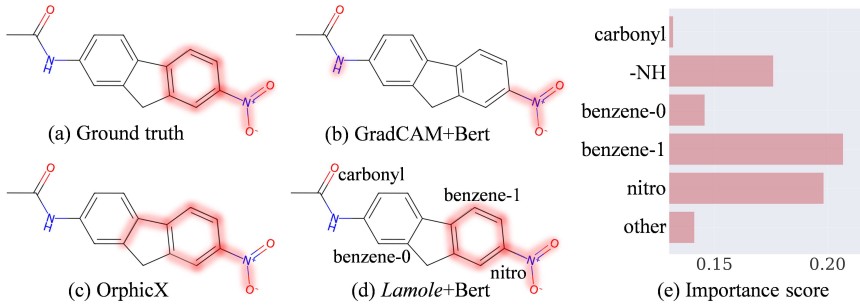

Figure 12: Explanation visualization of one molecule (ID: 156) from the Mutag dataset.

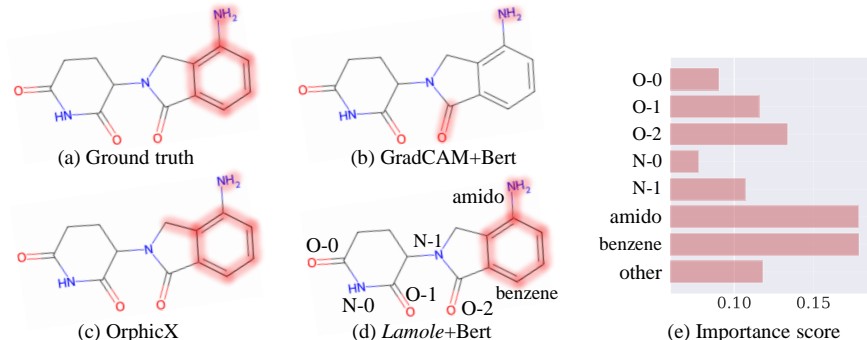

Figure 13: Explanation visualization of one molecule (ID: 574) from the Liver dataset.

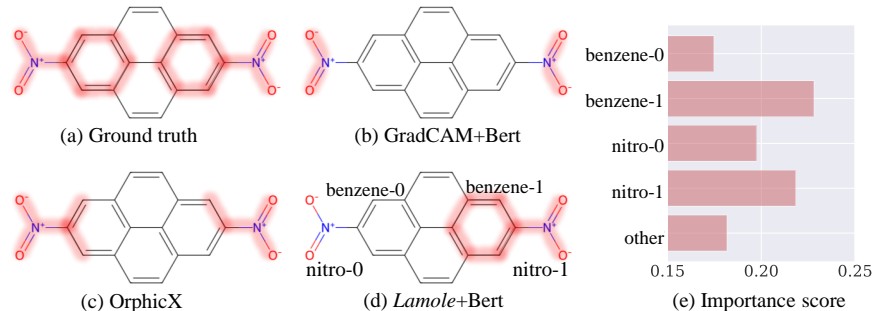

Figure 14: Explanation visualization of one molecule (ID: 161) from the PTC-FM dataset.

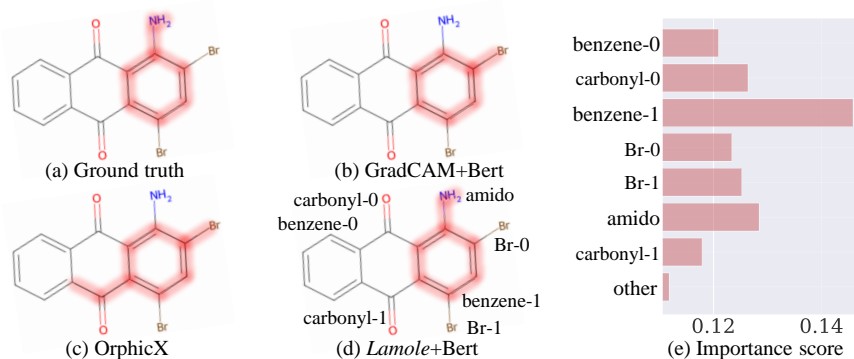

Figure 15: Explanation visualization of one molecule (ID: 277) from the PTC-FM dataset.

## A.8 COMPUTATIONAL COST

The pre-training stage took 11.8 hours. After pre-training, we fine-tuned the model on the used dataset. On the Mutag dataset, this process took 158s, and the evaluation time was 15s. For the baseline methods, SmoothGrad+GCN and OrphicX, the total training and evaluation time is 87s and 122s, respectively. Considering that the language model only requires pretraining once, the proposed method consumes acceptable additional computational cost but brings comparable classification accuracy and explainability.

## A.9 LIMITATION

The developed explainable molecular property prediction models have several limitations and require further research.

1. **Label annotations**: Currently, the proposed *Lamole* still requires human label annotations as additional supervisory signals. For a fair comparison, we also use these annotations to align the generated explanation of compared baselines. Table 2 shows that *Lamole* outperforms baseline explainability techniques when using human ground truth annotations. To eliminate the human labor in labeling annotations, we also explored the possibility of using LLMs to annotate ground truth, as shown in Appendix A.4. The results indicate that a few human label annotations are still required to improve the explanation accuracy. Despite that, from the ablation studies, we show that only a few annotations can significantly improve the explanation accuracy. Overall, there is a trade-off between the explanation accuracy and additional annotation costs.

2. **Generalizability**: Ensuring the generalizability of the explainable models to handle large and diverse molecular datasets across different chemical domains while maintaining interpretability and faithfulness to structure-property relationships is an ongoing challenge.

3. **Fidelity**: The classification prediction performance of the algorithm needs further improvement. Future work may include incorporating larger Group SELFIES corpora and larger models to further unleash its ability in explainable molecular property prediction.

