# OpenReview forum: "Explainable Molecular Property Prediction: Aligning Chemical Concepts with Predictions via Language Models"
_ICLR.cc/2025/Conference — ICLR 2025 Conference Withdrawn Submission_

### Official Review · Reviewer_6NMJ · 2024-10-27

**Soundness:** 1
**Presentation:** 2
**Contribution:** 2
**Rating:** 5
**Confidence:** 4

**Summary:**

The paper introduces Lamole, an explainable language-based model designed for predicting molecular properties. It uses Group SELFIES representation as input to a pre-trained language model and generates explanations using both gradients and attention weights. Additionally, it incorporates annotations from chemists to enhance the explanations provided.

**Strengths:**

1. The explainable molecular property prediction is crucial to drug discovery.
2. The paper is well-written and easy to follow.

**Weaknesses:**

1. My primary concern centers on the experimental section of the paper. The seven datasets employed are all relatively small, which may not fully demonstrate the scalability or generalizability of the proposed method. To address this, it would be beneficial to extend the testing to include medium-sized and large-sized datasets.
2. The experimental section would benefit from having more advanced baseline comparisons. Given that the proposed method uses Group SELFIES representation, it is essential to conduct a comparison with other motif-based graph explanation methods. Additionally, since the proposed method is a language-based model, it would be highly beneficial to compare it with other language-based models for molecular property prediction.
3. Explanation accuracy is insufficient for thoroughly evaluating an explanation method. Additional metrics, such as Fidelity +/-, should also be employed to assess the method's effectiveness.
4. The reliance on annotations that require domain knowledge presents a limitation, as most datasets do not include this information. This restricts the applicability of the proposed method across various datasets that lack domain-specific annotations.

**Questions:**

Please refer to Weaknesses.

---

### Official Review · Reviewer_E3LA · 2024-10-30

**Soundness:** 3
**Presentation:** 4
**Contribution:** 2
**Rating:** 6
**Confidence:** 4

**Summary:**

This paper introduces Lamole, an explainable molecular property prediction model based on language models. It employs GroupSELFIES, self-attention weights, and gradients to extract the impact of chemically meaningful substructures on the output of molecular property predictions.

**Strengths:**

- This work sheds light on the important but often overlooked topic of explainability in molecular property prediction.
- The comprehensive experiments, including effective visualizations such as attention scores and explanation visualizations, enhance the overall comprehensibility of the paper.

**Weaknesses:**

- The baselines presented in Table 1 are outdated, with the most recent one dating back to 2020. While I understand that the baselines for explanation accuracy may be limited, there is room for improvement in the property prediction baselines.
- Including a baseline with GROUPSELFIES (incorporating LSTM/Transformer) in Table 1 could strengthen the experimental results.

**Questions:**

- How is the threshold for the importance score determined? For example, in Figure 1(g), nitro-0 and benzene-1 do not appear to show a significant difference, as only nitro-0 and benzene-0 are presented as results.
- Why is the variance of the EP score much larger than that of the baselines?
- Do the results based on the experimental settings of GNNExplainer align with domain experts’ perspectives? Specifically, does high explanation accuracy truly indicate that the model explains well? The binary edge classification seems somewhat unconventional and lacks validity for me.
- Does the information flow-based explanation improve explainability in other domains? The approach appears to be general enough for broader applications.
- How is the ground truth defined in Figure 4?

---

### Official Review · Reviewer_HDdd · 2024-11-02

**Soundness:** 2
**Presentation:** 2
**Contribution:** 2
**Rating:** 5
**Confidence:** 4

**Summary:**

This work primarily focuses on providing meaningful and faithful explanations for molecular property predictions through the use of language models. To ensure scientifically meaningful explanations, Group SELFIES is employed to decompose SMILES strings into functional groups, which serve as basic explanatory units to maintain chemical relevance. Transformer-based LMs which are pretrained and fine-tuned with Group SELFIES strings, are leveraged to combine attention weights with gradients, capturing interactions between functional groups. Additionally, a marginal loss function and a theoretical analysis are introduced to enhance the explanation results. The effectiveness of the proposed approach is demonstrated through experiments on seven datasets, accompanied by visualizations that illustrate the outcome of the proposed method.

**Strengths:**

1) The paper incorporates language models (LMs) to facilitate scientifically meaningful explanations, potentially promoting advancements in related fields.

2) The paper proposes a novel method that combines attention weights with gradients to capture interactions between functional groups.

3) The paper presents a brief theoretical analysis that bridges the manifold hypothesis with explainable molecular property prediction.

4) The paper introduces a new model, Lamole, and demonstrates the effectiveness of the proposed methods through experiments on seven datasets. Ablation studies and visualizations further illustrate the performance of the proposed approach.

**Weaknesses:**

1) The novelty and contributions of this paper are limited. Although it aims to provide chemically meaningful explanations, understand functional group interactions, and faithfully reveal molecular structure-property relationships, it achieves little beyond addressing functional group interactions. Overall, the paper offers limited insights.

2) The proposed method employs gradients and attention mechanisms to capture functional group interactions; however, it lacks in-depth or theoretical analysis, despite the experimental results demonstrating the approach’s effectiveness.

3) The paper includes an excessive amount of information without clearly establishing connections to the primary research problem. For instance, on line 265, it states, “To address this issue, …” and subsequently introduces a max-margin loss function. However, it remains unclear which specific issue is being addressed and how this function directly relates to the problem. A similar lack of clarity appears on line 323, where the authors mention “projecting the causal features and spurious features into the data manifold.” It would be helpful to know if definitions of causal features and spurious features are provided in the paper. Furthermore, given that the paper focuses on identifying optimal subgraphs, it is unclear if these features are specifically discussed in the context of subgraphs.

4) The work relies heavily on functional groups obtained through Group SELFIES, which may limit its generalizability and broader applicability. While the primary objective of an explanation model is to offer user-friendly interpretations, the necessity of incorporating explanation methods diminishes if only a limited set of subgraphs is available.

5) Lastly, while the paper introduces LMs to address the problem, it provides little justification for their necessity or potential within this context.

**Questions:**

1) The presentation of ground truth explanations for the dataset is ambiguous. Line 826 states that “the ground truth for the mutagenic class is benzene with a chemical group, such as N=N, NO₂, and NH₂.” However, the limitations section mentions that “the proposed Lamole still requires human label annotations as additional supervisory signals.” This raises questions about how the ground truth is derived and the role of human annotations in this process.

2) If the ground truth annotations are human-derived, are they based on chemical principles or domain-specific knowledge? Additionally, are these ground truth explanations employed in the comparative methods, and if so, how are they implemented?

3) Providing a rationale for selecting baselines would strengthen the study. Given that the proposed method is an inherent explanation model, why are post-hoc methods chosen for comparison? Are there potentially more appropriate models available for this purpose?

4)  Regarding the chemical explanations, the claim in line 826 lacks persuasive evidence. Additional support from relevant chemical literature, such as [1], would enhance this assertion. Could the authors also provide further evidence from chemical sources to substantiate their claims?

5)  In Table 1, why does the GIN model outperform more complex models, such as ChemBERTa?

6) Are all datasets in this study composed of 2D graphs? What criteria are used for dataset selection, and could the six mutagenicity datasets be considered a single dataset due to their similarities? It would be beneficial to examine experimental results from other datasets.

7) Additional experimental results and conclusions from the ablation study would enhance the paper.

8) Given the limited number of functional groups in molecules, what advantages does an advanced architecture for explanations offer over simpler approaches like brute-force or heuristic search methods? It would be informative to see performance results on larger and more complex molecules.

9) In Section A.8 of the Appendix, the paper provides statistical data on training and inference times. What is the time complexity of the proposed method, and why are similar timing statistics not provided for all baseline methods?

10) How does the performance of the pre-trained models affect the final explanation results? Furthermore, why is the Zinc dataset selected for pre-training?

11) What advantages does explanation plausibility offer compared to explanation accuracy as an evaluation metric? Why is the title for Fig. 11 “Explanation Accuracy”?

12) Given that the performance of the baselines is similar, how are they trained or fine-tuned to ensure a fair comparison? Are techniques such as grid search employed? It would be helpful to provide further details.

[1]  Patlewicz, G., Rodford, R. and Walker, J.D., 2003. Quantitative structure‐activity relationships for predicting mutagenicity and carcinogenicity. Environmental Toxicology and Chemistry: An International Journal, 22(8), pp.1885-1893.

---

### Official Review · Reviewer_5dfj · 2024-11-03

**Soundness:** 2
**Presentation:** 2
**Contribution:** 2
**Rating:** 5
**Confidence:** 5

**Summary:**

The paper introduces Lamole, a framework designed for explainable molecular property prediction using transformer-based language models. Lamole utilizes Group SELFIES, a string-based molecular representation that encodes molecules at the functional group or fragment level, providing chemically meaningful tokens as input. The model integrates attention weights and gradients from the transformer architecture to compute importance scores for each substructure, aiming to generate explanations that align with chemical concepts. Additionally, a marginal loss function is introduced to align the model's explanations with chemists' annotations, enhancing the faithfulness of the explanations to the structure-property relationships. Experimental results on mutagenicity and hepatotoxicity datasets indicate that Lamole achieves comparable classification accuracy and improves explanation accuracy compared to baseline methods.

**Strengths:**

- The paper is well-organized and clearly written. It provides detailed explanations of the methodology, covering both theoretical foundations and practical implementations.

- Methodology-wise, it introduces a marginal loss function that aligns the model's explanations with chemists' annotations. This approach aims to improve the faithfulness of the explanations by ensuring they are grounded in expert knowledge, which is valuable for applications requiring interpretability.

- The paper conducts experiments on multiple datasets and compares the proposed method with several baselines, including both graph neural network (GNN)-based models and transformer-based models using SMILES representations. This comprehensive evaluation helps in assessing the performance of Lamole across different scenarios.

**Weaknesses:**

The technical novelty is limited, as Lamole is essentially a BERT model with Group SELFIES input and a new loss function. The use of attention weights combined with gradients for explanations lacks proper justification, especially given known issues with attention-based interpretability. The dependence on expert annotations severely limits practical applicability, while the marginal performance improvements don't justify the added complexity. The evaluation lacks comparisons with modern language models that could potentially generate explanations through prompting. The theoretical analysis of the marginal loss, while interesting, overshadows more practical concerns about model utility.

**Questions:**

1. Can you provide a summary of the major differences needed to be implemented in the BERT model to transform it into Lamole? Specifically, it appears that the main changes involve using Group SELFIES as input, adding a marginal loss function during training, and computing importance scores using attention weights and gradients. Are there any other architectural or substantial modifications required?

2. Can you include large language model (LLM)-based models in the comparison with Lamole, e.g. Llama 3.1 or ChemLLM[1]? In particular, we can use prompting to generate explanations for the most important parts of the molecule for LLM-based models.

3. How does the model scale with larger datasets and more complex molecules? Are there any computational bottlenecks introduced by the marginal loss or the computation of importance scores?

[1] Zhang, Di, et al. "Chemllm: A chemical large language model." arXiv preprint arXiv:2402.06852 (2024).

---

### Note · Authors · 2025-01-09

I have read and agree with the venue's withdrawal policy on behalf of myself and my co-authors.